# Research Progress and Direction of Novel Organelle—Migrasomes

**DOI:** 10.3390/cancers15010134

**Published:** 2022-12-26

**Authors:** Yu Zhang, Minghui Zhang, Zhuoyi Xie, Yubo Ding, Jialu Huang, Jingwei Yao, Yufan Lv, Jianhong Zuo

**Affiliations:** 1The Laboratory of Translational Medicine, Hengyang Medical School, University of South China, 28 Changsheng Road, Hengyang 421001, China; 2Nanhua Hospital, Hengyang Medical School, University of South China, Hengyang 421002, China; 3Clinical Laboratory, The Third Affiliated Hospital, Hengyang Medical School, University of South China, Hengyang 421900, China

**Keywords:** organelle, cell migration, migrasome, migracytosis, tumor

## Abstract

**Simple Summary:**

Migrasomes are a new type of organelle. Cells migrate and produce retraction fibers (RFs) with migrasomes on their backsides. When the RFs break, the migrasomes detach from the cell and the contents are taken up by surrounding cells. Migrasomes can integrate and transmit information, thereby regulating the physiological activities of the body. It has been confirmed that migrasomes are related to the occurrence and development of some diseases. In some cancers, the high expression of migrasomes is closely related to the occurrence and development of tumors. This review introduces the research progress of migrasomes and new ideas for research directions.

**Abstract:**

Migrasomes are organelles that are similar in structure to pomegranates, up to 3 μm in diameter, and contain small vesicles with a diameter of 50–100 nm. These membranous organelles grow at the intersections or tips of retracting fibers at the back of migrating cells. The process by which cells release migrasomes and their contents outside the cell is called migracytosis. The signal molecules are packaged in the migrasomes and released to the designated location by migrasomes to activate the surrounding cells. Finally, the migrasomes complete the entire process of information transmission. In this sense, migrasomes integrate time, space, and specific chemical information, which are essential for regulating physiological processes such as embryonic development and tumor invasion and migration. In this review, the current research progress of migrasomes, including the discovery of migrasomes and migracytosis, the structure of migrasomes, and the distribution and functions of migrasomes is discussed. The migratory marker protein TSPAN4 is highly expressed in various cancers and is associated with cancer invasion and migration. Therefore, there is still much research space for the pathogenesis of migratory bodies and cancer. This review also makes bold predictions and prospects for the research directions of the combination of migrasomes and clinical applications.

## 1. The Discovery of Migrasomes

The first prototype electron microscope, the transmission electron microscope (TEM), was invented in 1931 by German physicists Ernst Ruska and Max Knoll [1]. At that point, cell structures were beginning to be observed in greater detail. In 1945, Porter et al. observed filaments that may be related to the retraction edge of the cell [2]. Long tubular structures extending outward from the membrane of migrating cells were observed, and these structures were named retraction fibers (RFs) [3]. Few people studied this structure in the follow-up, until Professor Yu and his team discovered pomegranate-like structures (PLSs) through TEM in 2015 [4]. Oval-shaped membrane-bound structures outside the cells of the normal rat kidney (NRK) were found by TEM, which contained different numbers of vesicles. The diameter of these membrane structures was between 0.5 and 3 µm, and the structure was similar to an open pomegranate. Scanning electron microscopy (SEM) showed that these structures were attached to the RFs [4]. PLSs were isolated by density-gradient-centrifugation, and a variety of proteins that were enriched in the pomegranate body were quickly identified by mass spectrometry (MS) and green fluorescent protein (GFP) labeling methods. The PLSs that broke away from the cells destroyed and released the contents, and then the contents were absorbed by the surrounding cells. The formation of PLSs depends on cell migration, so PLSs are officially named “migrasomes”. The process by which cells release their contents includes cytosolic proteins. Among them, tetraspanin-4 (TSPAN4) is abundantly enriched in PLSs. Therefore, TSPAN4 serves as a marker protein for PLS [4]. Time-lapse imaging showed that when cells migrated, there were RFs behind them, and PLSs appeared at the intersections or tips of the RFs (Figure 1).

When the cells migrated further, the RFs continued to elongate until they broke. The PLSs broke away from the cells. PLSs destroyed and released the contents, and then the contents were absorbed by the surrounding cells. The formation of PLSs depends on cell migration, so PLSs are officially named “migrasomes” [4]. During random migration, cells not only move in a straight line, but also often change directions to varying degrees. Recent studies have confirmed that cells that begin to migrate have significantly fewer migrasomes than cells that migrate linearly. When cells migrate more persistently and faster, more migrasomes will be formed [5]. The process by which cells release their contents including cytosolic proteins and vesicles into the extracellular space through migrasomes is named “migracytosis” [4]. Their size, biogenesis process, and function are significantly different from those of exosomes (Table 1).

## 2. The Structure of Migrasomes

In 2015, migrasomes with PLSs with a diameter of approximately 0.5–3 μm outside the NRK cells were observed under TEM using an in situ embedding method. The number of small vesicles in them varies, up to 300 or less than 10 [4]. Observing the ultrastructures at the bottom of the cells, it was found that there were migrasomes at the intersection or top of the RFs. Further observation of the cell surface with field emission scanning electron microscopes (FE-SEM) showed that the RFs were connected with the migrasomes. Electron tomography was used to analyze the three-dimensional structure of the intersection of RFs and migrasomes on slices with a thickness of 250 nm. The results showed that some vesicles were fused at the intersection of the RFs, but the small vesicles in the migrasomes did not fuse with the outer membrane of the migrasomes [11]. High-pressure freezing and freeze substitution (HPF/FS) is the rapid freezing of samples under high pressure to prevent the formation of ice crystals and maintain the integrity of the tissue structure [12]. This method can observe dynamic changes in cells [13]. By observing the ultrastructure of the migrasomes through the HPF-FS sample preparation method, it was found that there were microfilaments, microtubules, and single vesicle transport in the RFs. Some migrasomes have fibrous structures, but not microfilaments. The plunging freezing sample preparation method can be used to further explore this structure in combination with cryogenic electron microscopy (Cryo-TEM) [11]. Cryo-TEM technology does not destroy the topological structure of the membrane. Through this technique, the cell structure at the subcellular level, the distribution of protein molecules, and the composition of some cytoskeletons can be observed, which is essential for understanding the formation and function of migrasomes [14].

## 3. The Molecular Mechanism of Migrasomes

### 3.1. Marker Protein—TSPAN4

TSPAN4 is enriched on migrasomes and can be used as a marker protein for migrasomes [4]. Neural crest cells (NCCs) can normally produce migrasomes without TSPAN overexpression [15]. Tetraspanins, also called transmembrane 4 superfamily (TM4SF) proteins, have four highly hydrophobic transmembrane domains [16]. There are currently 33 tetraspanins that are found in humans [17]. Tetraspanins play a role in physiological activities such as cell adhesion, activation, motility, and proliferation, and participate in pathological processes such as viral infection or metastasis [18,19]. Among tumor cells, TSPAN8 is involved in proliferation, metastasis, angiogenesis, and thrombosis [20,21]. CD151 plays a role in the regulation of integrin-dependent cell morphology and cell migration [22]. CD53 and CD37, which interact with immunoreceptors, are only expressed on immune cells [23]. By observing NRK epithelial cell lines that are stably expressing different levels of TSPAN4 and cell lines knocking out TSPAN4, it was confirmed that the expression of TSPAN4 promotes the generation of migrants, and the lack of TSPAN4 will reduce the generation of migrants in the cells [24]. Tetraspanin proteins and cholesterol form Tetraspanin-enriched microdomains (TEMs) on the membranes [17]. Studies have confirmed that the migrasomal membrane is not only rich in TSPAN4 and cholesterol, but also contains integrins and other transmembrane proteins. These micron-scale macrodomains are named TEMAs. Migrasomes are formed due to the presence of TEMAs that swell into large vesicular migrasomal shapes. An in vitro membrane system was designed to simulate the formation of RFs and migrasomes. They are giant unilamellar vesicles (GUVs) containing purified TSPAN4, cholesterol, and other lipids. The results show that GUVs containing TSPAN4 and cholesterol can form migrasome-like structures, while GUVs without cholesterol or TSPAN4 cannot form a migrasome-like structure. Another device also obtained the same result. To explain the physical mechanism of migrasome formation, Professor Yu’s research group and Professor Kozlov’s research group collaborated to establish a theoretical model [24]. The results confirmed that TSPAN4 protein and cholesterol were locally enriched on RFs due to cell migration, which increased the bending rigidity of TEMAs, thereby forming migrasome-like structures [24]. Migrasomes contain CD63 [7]. They were observed in NCCs expressing low levels of pCMV-CD63-pHluorin (CD63-pH) and were stained with BODIPY ceramide. The results showed that RFs were particularly enriched in CD63-pH and migrasomes were brightly labeled with BODIPY ceramide. This suggests that BODIPY ceramide could serve as a migrasome marker [15]. TSPAN4 has pan-cancer significance in most cancer types according to bioinformatics analysis [25]. The expression of TSPAN4 in gastric cancer tissues was significantly higher than that in paracancerous tissues. The downregulation of TSPAN4 in tumor xenografts was able to suppress tumor formation, suggesting that this gene may have a retarding effect on gastric cancer progression [26]. TSPAN4 is highly expressed in lung adenocarcinoma (LUAD) and promotes the metastasis of LUAD [27]. Therefore, TSPAN4 may be a biomarker and a potential therapeutic target for some cancers.

### 3.2. Integrin Mediation

Integrins are transmembrane receptors that mediate the connection between cells and the extracellular matrix (ECM) [28]. Integrins not only mediate signals from the outside to the inside, but also mediate cellular signals from the inside to the outside [29]. Integrins are obligate heterodimers that are composed of α and β subunits [30]. In mammals, a total of 18 alpha subunits and 8 beta subunits have been found [31]. The results of mass spectrometry showed that integrin α5β1 was enriched on migrasomes. Further studies have shown that β1 localized by the migratory body is in its activated ligand binding state, integrin α5-GFP is highly enriched in the migratory body, and integrin α5-GFP on the retracting fiber is relatively small. In addition, three-dimensional imaging showed that the integrins were located at the bottom of the migrasomes. During the formation of migrasomes, the integrins on the cell bodies enable cells to migrate, and the integrins on the migrasomes provide adhesion for retraction fiber tethering [32].

### 3.3. ROCK1 Regulates Cell Adhesion

ROCK1 is a regulator of migrasome formation, as determined by chemical screening [33]. By constructing NRK cells that were stably expressing TSPAN4-GFP, the migrasomes that were produced by the cells were treated with the compound. After multiple screenings of 2240 compounds, it was shown that SAR407899 can stably inhibit the formation of migrasomes. In zebrafish embryos, the production of migrasomes can be observed during gastrulation which confirmed that the formation of migrasomes was very important for organ morphogenesis during embryonic development [34]. In zebrafish embryos that were treated with SAR407899, the number of migrasomes was reduced, and the formation of Kupffer’s vesicles (KVs) was also significantly disrupted. SAR407899 is an inhibitor of ROCK2 and ROCK1. Knockdown of ROCK2 does not interfere with the biogenesis of migrasomes, while the amount of migrasomes that are produced by ROCK1 knockout cells is significantly reduced. The knockdown of ROCK1 affects cell migration [35]. ROCK1 plays a role in cell adhesion to fibronectin [36]. The number of migrasomes increased with the concentration of fibronectin in WT cells, suggesting that their formation is dependent on cell adhesion to fibronectin. However, the number of migrasomes did not change significantly with the concentration of fibronectin in ROCK1 knockdown cells due to impaired cell adhesion to fibronectin. It was observed by traction force microscopy that the knockdown ROCK1 cells generated significantly less traction force than WT cells. It was further confirmed that ROCK1 regulates the formation of migrasomes by attaching to fibronectin to generate traction [33].

## 4. Physiological Function of Migrasomes

### 4.1. Deliver, Integrate, and Release Information

Zebrafish embryos have the characteristics of rapid growth, are large and transparent, and can develop in vitro [37,38]. Zebrafish embryos were selected as a model system to observe whether migrasomes participate in biological processes in vivo. Through fluorescence labeling and time-lapse imaging analysis, vesicles that were enriched in TSPAN4 and integrin β1 and long projections resembling retraction fibers were observed in shield-stage zebrafish embryos. Furthermore, combined with TEM observations, it was confirmed that migrasomes were produced during zebrafish gastrulation. To determine the distribution of migrasomes in zebrafish embryos, the migrasomes were marked by overexpressing pleckstrin homology-green fluorescent protein (PH–GFP). The number of migrasomes increased significantly at 5.5 h post fertilization (h.p.f.) until it peaked at 7 h. p. f. A large number of migrasomes are produced by mesodermal cells and endodermal cells during gastrulation. These migrasomes are enriched in a layer of specific structures that are located between the endoderm and the yolk syncytial layer in the zebrafish embryo [34].

Integrins can promote cell adhesion and migration [39]. By constructing a genetic model of the defect in the production of migrasomes, it was confirmed that TSPAN4a and TSPAN7 and integrin β1b regulate the production of migrasomes by providing adhesion in early zebrafish embryos. When the production of migrasomes is blocked, zebrafish will have abnormal organ phenotypes. These phenotypes include organ morphological deformation, volume reduction, and left-right overturning or duplication of organs. By supplementing exogenous migrasomes, the proportion of embryo defects is significantly reduced. 

Quantitative mass spectrometry analysis revealed that migrasomes are rich in 18 kinds of signaling molecules such as Tgfβ2, PdgfD, Cxcl12b, Cxcl12a, Wnt8a, Bmp7a, Cxcl18a, Wnt5b, and Bmp2. A lack of Cxcl12 will cause changes in organ morphology [40]. Further experiments confirmed that the migrasomes produce Cxcl12a, thereby affecting organ morphogenesis through the Cxcl12a-Cxcr4b signal axis [41]. In these morphological defects, the team selected the laterality with a more mature mechanism for analysis. Kupffer’s vesicle (KV) in zebrafish is involved in breaking the left–right (LR) symmetry in the body [42]. TSPAN4a and TSPAN7 participate in KV formation. During gastrulation, dorsal forerunner cells (DFCs) migrate to the front edge of the zebrafish embryonic shield and then form KV [43,44]. DFC cells lacking TSPAN4 and TSPAN7 at the proembryonic stage cannot gather tightly on the front edge of the embryonic shield during migration. Rapid three-dimensional (3D) imaging showed that there is a cavity under the DFCs. This cavity was named the embryonic shield cavity and it was determined that migrasomes accumulate in the embryonic shield cavity [34]. Through the in vivo chemoattraction assay, it was confirmed that the migrasomes are chemoattractants for DFC [34]. With the development of fluorescent artificial antigens (FAAs), a network of membrane fibers of dendritic cells (DCs) for antigen uptake can be observed. This confirms that cytoplasmic components of DCs, such as chemokines and antigens, may be transferred to other DCs via migrasomes for intercellular communication [45]. In summary, migrasomes are enriched with signaling molecules, including chemokines, cytokines, morphogens, and growth factors. These signaling molecules can be transported to specific locations and simultaneously released from the migrasomes. In this way, the migrasomes can regulate the correct spatial positioning and development of each organ in the embryo.

### 4.2. Mediate the Lateral Transfer of mRNA and Protein

Migrasomes can be phagocytosed by surrounding cells to achieve material transport and information transmission between cells [4]. Research has confirmed that migrasomes can be stained by SYTO 14 fluorescence [7]. SYTO 14 is stained only by RNA fluorescence [46]. Approximately 30% of migrasomes contain RNA, and the degree of RNA enrichment in migrasomes varies greatly. Researchers isolated purified migrasomes and small EVs from L929 cells for control experiments. In migrasomes, the length distribution of RNA is dominated by long RNA species, and RNA is mainly composed of mRNA species. Pten, which is abundant in migrasomes, was selected for analysis. Pten is a tumor suppressor gene [47]. It was found that Pten mRNA can be transferred to recipient cells with migrasomes, reducing the pAKT activity of the recipient cells and finally inhibiting the proliferation of the recipient cells [7]. In summary, migrasomes can mediate the lateral transfer of mRNA and protein, carry out information transmission, and realize physiological functions.

### 4.3. Homeostasis Regulation of Mitochondria

Mitochondria are double-membrane organelles in eukaryotic cells [48]. In addition to providing energy for cells, mitochondria also regulate cell growth, differentiation, and apoptosis [49]. Mitophagy, the mitochondrial proteasome, the ubiquitin-proteasome system (UPS), and the degradation of mitochondrial-derived vesicles (MDVs) are involved in the regulation of mitochondrial quality [50,51]. The cells were treated with mitochondrial stressors, and mitochondria were found to gather in the migrasomes of the migrating cells, and were discharged from the cells with the migrasomes. This process is named mitocytosis, and this phenomenon exists in many kinds of cells. The mechanism was further studied and it was found that during mitocytosis, the tubular mitochondria extended close to the plasma membrane and the tips became fragments. As the cells migrated, the fragments entered the migrasomes. These mitochondria have abnormal morphology and are damaged mitochondria. The gene knockout experiment further explored the influencing factors of mitochondrial movement. The kinesin family 5B gene (KIF5B) can drive mitochondria to move outward [52]. Mitofusin-1 (MFN1) and mitofusin-2 (MFN2) can promote mitochondrial outer membrane fusion [53]. Dynein-related protein 1 (DRP1) is a key factor in mitochondrial fission [54]. Myosin-19 (Myo19) can bind to mitochondria and actin and mediate shorter-range mitochondrial movement in the cell [55]. Knockout experiments were separately performed on the above genes in cells. The results showed that KIF5B promoted the elongation of tubular mitochondria to the plasma membrane, Myo19 tightly connected mitochondria and actin, and drp1 ruptured the tips of mitochondria into fragments and entered the migrasome. Most of the mitochondria that are located at the bottom of the cells and in the migrasomes are MitoSOX positive and TMRM negative [6]. MitoSOX can detect mitochondrial reactive oxygen species (ROS) levels. ROS are produced in the mitochondria and cause mitochondrial dysfunction and ultimately damage the body [56]. The fluorescent dye tetramethylrhodamine methyl ester (TMRM) can reflect the mitochondrial membrane potential [57]. These results indicate that the mitochondria in the migrasomes are damaged, and the damage occurs before mitocytosis. By constructing a cell model of mitochondrial heteroplasmy, it was confirmed that mitochondria with a deleterious mutation in the mtDNA could be selectively disposed of by mitocytosis [6,58]. Damaged mitochondria release signals, bind to the outward motor protein KIF5B, and move to the outside of the cell. Further studies have suggested that mitocytosis occurs in macrophages to maintain mitochondrial quality regulation. Neutrophils that exist in large numbers in the body are highly migratory [59], and the detection of migrasomes shows that the high circulation of neutrophils is inseparable from the support of mitocytosis. Mitocytosis is a new mitochondrial quality control mechanism that is mediated by migrasomes in the cell. 

## 5. Migrasomes in the Diagnosis and Treatment of Disease

It has been confirmed that migrasomes are very likely to be used as markers for disease diagnosis and prognosis in the fields of cardiovascular and cerebrovascular diseases, kidney disease, and bone-related diseases (Figure 2). Migrasomes are involved in the sodium chloride-driven pathogenesis of ischemic stroke disease. By constructing a mouse model of ischemic stroke that is induced by a high-salt diet, it was suggested that a high-salt diet triggered the formation of migrasomes during cerebral ischemia [60]. It is known that sodium chloride can promote the production of inflammatory factors by T-cells and macrophages and aggravate cell damage [61]. It was experimentally demonstrated that sodium chloride induces microglia to form migrasomes and leads to a pro-inflammatory polarization of microglia [60].

Kidney diseases are often accompanied by the appearance of proteinuria [62]. In the kidney, podocytes form the filtration barrier of the glomerulus. Their injury leads to an increase in glomerular permeability and the development of proteinuria [63]. Therefore, detecting podocyte damage is the key to the diagnosis and treatment of glomerular diseases. In 2020, to study the relationship between the degree of podocyte damage and the urine level of migrasomes that were released by podocytes, researchers established a kidney injury mouse model that was induced by puromycin amino nucleoside (PAN), LPS, or a high concentration of glucose (HG). They also further analyzed urine samples from patients with active diabetic nephropathy (DN) and detected migrasomes in the urine. The results confirmed that podocytes can release migrasomes, and the number of migrasomes that are released is proportional to the degree of damage to the podocytes. The inhibition of podocyte RAC-1 activity can significantly reduce the number of migrasomes in urine [64]. Rac-1 is a gene that is related to migration and inflammation and inhibiting its activity can protect the function and structure of podocytes [65,66]. By detecting the source of migrasomes in urine, it was found that most of the migrasomes were derived from podocytes [64]. Podocytes have high mobility [67]. Healthy podocytes are in a static state, and damaged podocytes will migrate [68]. The increased number of urinary migrasomes was detected earlier than the elevated proteinuria [64]. Therefore, migrasomes in urine that depend on cell movement can be used as an ideal indicator for the diagnosis of early podocyte damage.

The development and remodeling of bone are inseparable from osteoclasts (OCs) [69]. This process refers to information transmission and identification between adjacent cells. The mononuclear pre-OCs migrate to other nearby mononuclear pre-OCs and fuse to form multinucleated cells, thereby generating mature OCs [70]. By using wheat germ agglutinin (WGA)-fluorescein isothiocyanate (FITC)-labeled RAW 264.7 cells to study the mechanism of OC formation, it was discovered that OCs produce migrasomes [71,72]. Therefore, it was proposed that the function of the migrasomes was to serve as information carriers, and the cells that migrate first would leave a message behind to point out the direction for other cells [72]. This study provides a new idea for the mechanism of osteoclast migration and fusion, which is of great significance to bone-related diseases such as osteoporosis and rheumatoid arthritis.

Migrasomes have the function of information transmission. They release the cell contents into the extracellular environment through migracytosis, and can also be taken up by neighboring cells to transmit biochemical information and location information. In terms of neural network formation and immune response, the transmission of location information is particularly important, so migrasomes may play a huge role in these two aspects [73,74].

Peptides are a new direction of regenerative medicine. Peptide interfaces containing cell-penetrating peptides (pVECs) were found to promote cell migration and migrasome formation using primary normal human dermal fibroblast (NHDF) studies [75]. Fibroblasts can be used for wound healing studies and dermatological research to investigate diseases such as fibrosarcoma, fibrosis, histiocytoma, and xeroderma pigmentosum [76,77]. Migrasomes will provide new ideas for cancer research, tissue regeneration, and tissue engineering research.

## 6. Conclusions and Perspective

Research on migrasomes needs further exploration. It is currently known that migrasomes are monolayer membrane organelles containing many small vesicles [4]. It is still unclear where these vesicles originated from, how they were formed, and the contents of the migrasomes. The mechanism by which they are specifically sorted and transported is also unclear. Finding the marker proteins of these vesicles will help in studying the source of small vesicles in the migrasomes and the formation of the migrasomes. 

Migrasomes have the function of information transmission. They release the cell contents into the extracellular environment through migracytosis and can also be taken up by neighboring cells to transmit biochemical information and location information. In terms of neural network formation and immune response, the transmission of location information is particularly important, so migrasomes may play a huge role in these two aspects [73,74].

The generation of migrasomes depends on cell migration [78]. In the body, cell migration is involved in processes such as embryonic development, immune response, angiogenesis, inflammatory response, wound healing, and cancer metastasis [79,80,81,82]. Relevant studies have confirmed that viruses can spread by inducing the formation of migratory bodies to promote cell migration [83]. This provides a new way of thinking for virus research.

Therefore, it is still necessary to explore whether migrasomes play a role in these physiological activities, and what role they play. Some of them have preliminary verification, but further research is needed. Researchers can build simple, easy-to-observe biological models that are similar to zebrafish, and study the important role of migrasomes in biological development. Integrin and rock1 mediate the adhesion of migrasomes to the extracellular matrix [32]. Cell adhesion supports the occurrence and development of cancer [84,85]. Therefore, whether the generation of migrasomes is related to the immune escape and metastasis of cancer should be considered. If this hypothesis is true, migrasomes may become a marker for disease diagnosis, an indicator of disease prognosis, and a standard for cancer classification in the future. The tumor microenvironment is currently a hot research topic. Tumor tissues are mostly in a hypoxic environment [86]. There is no relevant research on the relationship between the hypoxic environment and migrasomes. Therefore, it is still necessary to further study whether there is a difference between migrasomes under hypoxic and normoxic conditions and what the difference is. 

With the development of technology and the deepening of multidisciplinary cooperation, research on the mechanisms of the generation, transfer, and release of migrasomes will become increasingly thorough. Researchers can combine research results with clinical cases to realize the conversion of the theoretical knowledge of migrasomes into the diagnosis and treatment of diseases.

## Figures and Tables

**Figure 1 cancers-15-00134-f001:**
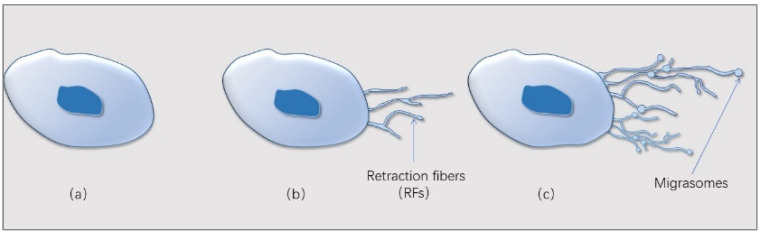
The production process of migrasomes. (**a**) Cells migrate, (**b**) retraction fibers (RFs) are produced at the back of the cells, and (**c**) migrasomes grow at the ends or junctions of the RFs. When RFs break, the migrasomes break away from the cells, and finally break. The contents of the migrasomes are taken up by the surrounding cells.

**Figure 2 cancers-15-00134-f002:**
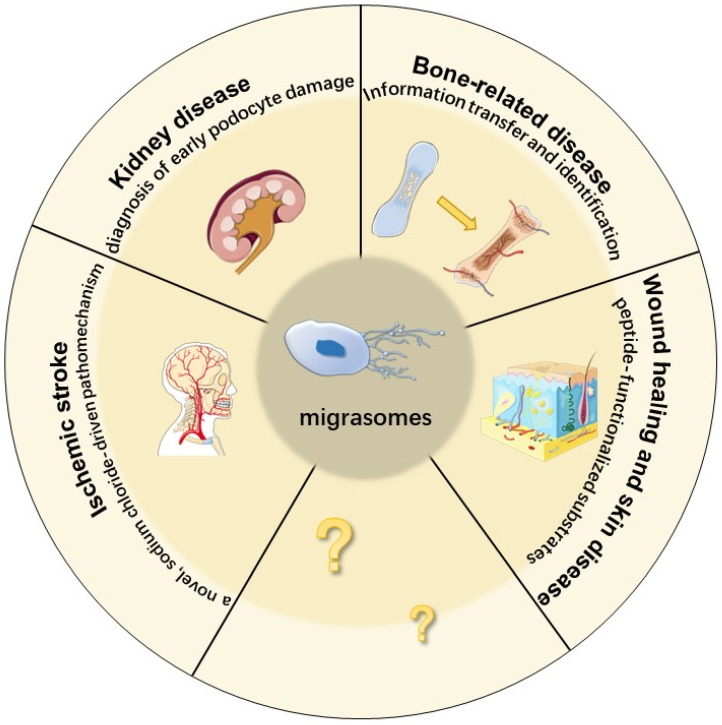
Migrasomes and related diseases.

**Table 1 cancers-15-00134-t001:** Comparison of migrasomes and exosomes.

	Migrasomes	Exosomes
Diameter	500–3000 nm	30–150 nm
Biogenesis process	Formed by the assembly of large domains on the migrating cytoplasmic membrane	Multivesicular bodies (MVBs) generate vesicles and fuse with the plasma membrane to release exosomes
Essence	Organelle	Extracellular vesicles (EVs)
Function	Regulates organ development, mediates intercellular mRNA transmission, and maintains intracellular mitochondrial homeostasis	Antigen presentation, tumor growth and migration, tissue damage repair
References	[4,6,7]	[8,9,10]

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
