# Peer review of "Research Progress and Direction of Novel Organelle—Migrasomes"

_cancers, 2022, doi:10.3390/cancers15010134_

Round 1

Reviewer 1 Report

This is a fascinating overview of the novel migrasome research field. The possible implications of this research interaction can be outstandingly important for both physiological and pathological processes.

Concerns

First of all, even though the review subject is very interesting it is not directly correlated to cancer research. The field is in its beginning and this should be discussed in the abstract and introduction section.

The title could be rephrased.

The English style and grammar is rather poor and needs to be extensively corrected throughout the manuscript as it makes comprehension difficult at specific points.

E.g line 62 "turning cells"

line 37 "humans"

lines 53 and 92, typographic errors

line 120 change" Tel Aviv University" to the more appropriate reference

line 151 please elaborate on the fibronectin sentence, including reference

line 166 please modify the sentence

lines 168 and 174 and throughout the manuscript, please change paragraphs

line 252, please change confirmed to suggested

line 311 sentence paragraph

A better organization of the perspective with a figure illustrating possible migrasome roles would benefit the manuscript

Author Response

  1. It is not directly correlated to cancer research. The field is in its beginning and this should be discussed in the abstract and introduction section.

I added a discussion of migrasomes and cancer in the abstract and introduction sections.

  1. The title could be rephrased.

The title has been changed to Research progress and direction of novel organelle—microsomes

  1. The English style and grammar is rather poor and needs to be extensively corrected throughout the manuscript as it makes comprehension difficult at specific points.

The article has been extensively revised in English.

  1. g line 62 "turning cells" —— During random migration, cells not only move in a straight line, but also often change directions to varying degrees.

line 37 "humans"——Cell structures were beginning to be understood in greater detail.

lines 53 and 92, typographic errors——The formation of PLSs depends on cell migration, so PLSs are officially named “migrasomes”. The process by which cells release their contents includes cytosolic proteins. Among them, tetraspanin-4 (TSPAN4) is abundantly enriched in PLSs.

The Plunging freezing sample preparation method can be used to further explore this structure in combination with cryogenic electron microscopy (Cryo-TEM)[11]. Cryo-TEM technology does not destroy the topological structure of the membrane.

line 120 change" Tel Aviv University" to the more appropriate reference——Prof. Kozlov’s

line 151 please elaborate on the fibronectin sentence, including reference——ROCK1 plays a role in cell adhesion to fibronectin[36]. The number of migrasomes increased with the concentration of fibronectin in WT cells, suggesting that their formation is dependent on cell adhesion to fibronectin. However, the number of migrasomes did not change significantly with the concentration of fibronectin in ROCK1 knockdown cells due to impaired cell adhesion to fibronectin. It was observed by traction force microscopy that knockdown ROCK1 cells generated significantly less traction force than WT cells. It was further confirmed that ROCK1 regulates the formation of migrasomes by attaching to fibronectin to generate traction [33].

line 166 please modify the sentence——The number of migrasomes increased significantly at 5.5 h post fertilization (h.p.f.) un-til it peaked at 7 h. p. f.

lines 168 and 174 and throughout the manuscript, please change paragraphs——Modifications have been made

line 252, please change confirmed to suggested——Further studies have suggested that mitocytosis occurs in macrophages to maintain mitochondrial quality regulation.

line 311 sentence paragraph——Modifications have been made

Reviewer 2 Report

The work is clear and well organised. It is original and can give a good contribution to consolidate the literature in the topic of migrasome.

One observation: the figure should be changed: avoid the word migrasome in the centre and use a draw, make bigger the other organs and be more specific in which way the migrasome can contribute to the disease at least with two words.

Minor observation. TSPN4 is not always written in capital.

Please check also spelling along the text.

Reviewer 3 Report

in this manuscript Yu zhang and co-authors provide the reader with information about a not-very-well-known cellular structure: the migrasome.
Overall, the topic is interesting, but the entire manuscript requires exstensive language revision, and many concepts must be better presented or clarified to proceed with a more accurate review
A complete revision is needed, some examples are reported below:

lines:
54; 57; 65; 109;
114 (in vitro membrane system);
122 (only the term TEMs was described before);
124 (probably pHluorin conjugated CD63);
150 (reformulate and move upwards in the text) the whole paragraphe is too condensed.

Author Response

  1. The entire manuscript requires extensive language revision, and many concepts must be better presented or clarified to proceed with a more accurate review

A complete revision is needed, some examples are reported below:

The article has been extensively revised in English.

  1. lines:

54; The process by which cells release their contents includes cytosolic proteins. Among them, tetraspanin-4 (TSPAN4) is abundantly enriched in PLSs.

57; When the cells migrated further, the RFs continued to elongate until they broke. The PLSs broke away from the cells.

65; The process by which cells release their contents including cytosolic proteins and vesicles, into the extracellular space through migrasomes is named " migracytosis "[4]. Their size, biogenesis process, and function are significantly different from those of exosomes(Table 1).

109; By observing NRK epithelial cell lines stably expressing different levels of TSPAN4TSPAN4 and cell lines knocking out TSPAN4TSPAN4, it was confirmed that the expression of TSPAN4TSPAN4 promotes the generation of migrants, and the lack of TSPAN4TSPAN4 will reduce the generation of migrants in the cells[24].

114;  (in vitro membrane system);

An in vitro membrane system An extracorporeal membrane device was designed to simulate the formation of RFs and migrasomes.

122;  (only the term TEMs was described before);

      These micron-scale macrodomains  are named TEMAs. Migrasomes are formed due to the presence of TEMAs that swell into large vesicular migrasomal shapes. An in vitro membrane system was designed to simulate the formation of RFs and migrasomes.

124;  (probably pHluorin conjugated CD63);

     They were observed in NCCs expressing low levels of pCMV-CD63-pHluorin (CD63-pH) and stained with BODIPY ceramide.

150;  (reformulate and move upwards in the text) the whole paragraph is too condensed.

     Knockdown of ROCK2 does not interfere with the biogenesis of migrasomes, while the amount of migrasomes produced by ROCK1 knockout cells is significantly reduced. Knockdown of ROCK1 affects cell migration[35]. ROCK1 plays a role in cell adhesion to fibronectin[36]. The number of migrasomes increased with the concentration of fibronectin in WT cells, suggesting that their formation is dependent on cell adhesion to fibronectin. However, the number of migrasomes did not change significantly with the concentration of fibronectin in ROCK1 knockdown cells due to impaired cell adhesion to fibronectin. It was observed by traction force microscopy that knockdown ROCK1 cells generated significantly less traction force than WT cells. It was further confirmed that ROCK1 regulates the formation of migrasomes by attaching to fibronectin to generate traction[33].

Round 2

Reviewer 1 Report

The authors have responded to concerns.